# Osmotic Demyelination Syndrome: A Rare Clinical Image

**DOI:** 10.3390/diagnostics13213393

**Published:** 2023-11-06

**Authors:** Prishita Koul, Pallavi Harjpal, Raghuveer Raghumahanti

**Affiliations:** Department of Neurophysiotherapy, Ravi Nair Physiotherapy College, Datta Meghe Institute of Higher Education and Research, Wardha 442107, India; pallavi.harjpal@dmiher.edu.in (P.H.); raghuneuro@yahoo.com (R.R.)

**Keywords:** osmotic demyelination syndrome (OSD), hyponatremia, pons, hyperglycaemia

## Abstract

The term “Osmotic Demyelination Syndrome” (ODS) is synonymous with central pontine myelinolysis (CPM), denoting a condition characterised by brain damage, particularly affecting the white matter tracts of the pontine region. This damage arises due to the rapid correction of metabolic imbalances, primarily cases of hyponatremia. Noteworthy triggers encompass severe burns, liver transplantations, anorexia nervosa, hyperemesis gravidarum, and hyperglycaemia, all linked to the development of CPM. Clinical manifestations encompass a spectrum of signs and symptoms, including dysphagia, dysarthria, spastic quadriparesis, pseudobulbar paralysis, ataxia, lethargy, tremors, disorientation, catatonia, and, in severe instances, locked-in syndrome and coma. A recent case involving a 45-year-old woman illustrates these complexities. Upon admission to the Medicine Intensive Care Unit, she presented with symptoms indicative of diminished responsiveness and bilateral weakness in the upper and lower limbs. Of significance, the patient had a pre-existing medical history of hyperthyroidism. Extensive diagnostic investigations were undertaken, revealing compelling evidence of profound hyponatremia through blood analyses. Furthermore, magnetic resonance imaging (MRI) was performed, unveiling conspicuous areas of abnormal hyperintensity located in the central pons, intriguingly accompanied by spared peripheral regions. These radiological findings align with the characteristic pattern associated with osmotic demyelination syndrome, illuminating the underlying pathology.

The following magnetic resonance imaging (MRI) images displaying different indicators observed in osmotic demyelination syndrome (Figure 1).

## Figures and Tables

**Figure 1 diagnostics-13-03393-f001:**
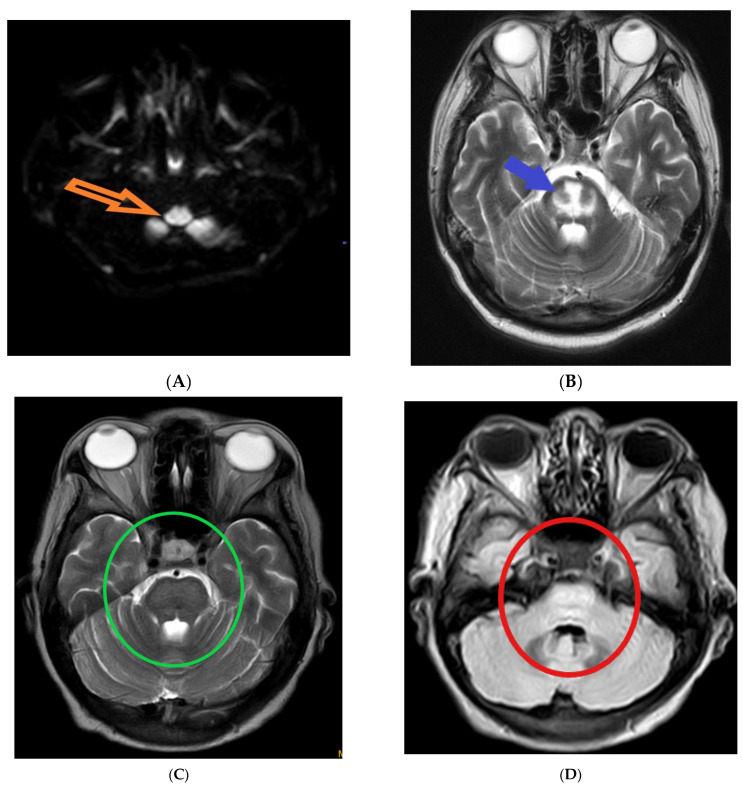
Magnetic resonance imaging: (**A**) hyper-intensity of central pons in diffuse weighted image (orange arrow) [1]; (**B**) trident-shaped appearance (omega sign) of central pons in T2 weighted image (blue arrow) [2]. The characteristic “trident”-shaped appearance is attributed to the primary affectation of the transverse pontine fibres and the relatively limited involvement of the descending corticospinal tracts [3]; (**C**,**D**) piglet sign appearance of upper pons in T2 and FLAIR images (green and red circles, respectively) [4]. T2-weighted MR images exhibit a distinctive pattern known as the "piglet face" sign, initially described by Wagner et al. [5]. In this sign, the pons takes on a distinctive resemblance to the snout of a piglet, while the internal carotid arteries (ICAs) and the fourth ventricle together form the eyes and mouth, respectively, of the piglet-like configuration.

## Data Availability

Not applicable.

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
