# Peer review of "Osmotic Demyelination Syndrome: A Rare Clinical Image"

_diagnostics, 2023, doi:10.3390/diagnostics13213393_

Round 1

Reviewer 1 Report

In this article, MR images of a patient with central pontine myelinolysis (osmotic demyelination syndrome) are presented. Although the subject is rare and intriguing, the text and visual material require more detail.

1)      It is necessary to add an explanation about Figure 1A

2)      The Omega sign (Trident-shaped appearance) that is defined in Figure 1B is not visible.

3)      Typical MR findings of osmotic demyelination should be added briefly. It is possible to provide a brief overview of the Omega sign and piglet sign's significance, as well as its origin.

Author Response

Dear Reviewer,

"I've implemented the recommended changes you mentioned earlier and have included them in the attached file. I appreciate your assistance in reviewing the manuscript."

1) It is necessary to add an explanation about Figure 1A:

Ans: Hyper-intensity of central pons in diffuse weighted image (orange arrow)

2)      The Omega sign (Trident-shaped appearance) that is defined in Figure 1B is not visible.

Ans:  I have done the changes.

3) Typical MR findings of osmotic demyelination should be added briefly. It is possible to provide a brief overview of the Omega sign and piglet sign's significance, as well as its origin.

Ans: Trident-shaped appearance (omega sign) of central pons in T2 weighted image (blue arrow) [2]. The characteristic "trident" shaped appearance is attributed to the primary affectation of the transverse pontine fibers and the relatively limited involvement of the descending corticospinal tracts [3] ; (C), (D) Piglet sign appearance of upper pons in T2 and FLAIR images (green and red circles respectively) [4]. T2-weighted MR images exhibit a distinctive pattern known as the "piglet face" sign, initially described by Wagner et al. [5]. In this sign, the pons takes on a distinctive resemblance to the snout of a piglet, while the internal carotid arteries (ICA) and the fourth ventricle together form the eyes and mouth, respectively, of the piglet-like configuration.

Reviewer 2 Report

I would like to congratulate the authors for an excellent manuscript on a rare clinical scenario along with helpful images.

I would encourage authors to provide more clinical information and outcomes. 

Also if the authors can provide in concise  form about the uniqueness of this case as compared to others would be helpful

Author Response

Dear Reviewer,

"I've implemented the recommended changes you mentioned earlier and have included them in the attached file. I appreciate your assistance in reviewing the manuscript."

Thank You

Reviewer 3 Report

This paper needs to be reorganized. The authors need to correct the points that are emphasized below.

Point 1. There needs to be a description of part A in Figure 1. Also, the legend does not show what the orange arrow represents. The legend must be more detailed.

Point 2. Reference 1 needs to be included in the text below the Figure 1. It needs to be cited.

Minor editing of English language required

Author Response

(The authors gave the same response as above.)

Round 2

Reviewer 1 Report

I think the manuscript is now suitable for publication.

Reviewer 3 Report

The authors corrected the paper according to the suggestions.